# Chronic exercise modulates the cellular immunity and its cannabinoid receptors expression

Salvador Valencia-Sánchez[1], Karen Elizabeth Nava-Castro[2ᴼ], Margarita Isabel Palacios-Arreola[2ᴼ], Oscar Prospéro-García[3‡], Jorge Morales-Montor[4]*, René Drucker-Colín[1†‡]

**1** Instituto de Fisiología Celular, Departamento de Neuropatología Molecular, Universidad Nacional Autónoma de México, Circuito Exterior s/n, Ciudad Universitaria, Ciudad de México, México,
**2** Genotoxicología y Mutagénesis Ambientales, Ciencias Ambientales, Centro de Ciencias de la Atmósfera, Ciudad de México, México, **3** Laboratorio de Cannabinoides, Departamento de Fisiología, Facultad De Medicina, Universidad Nacional Autónoma de México, Circuito Exterior s/n, Ciudad Universitaria, Ciudad de México, México, **4** Departamento de Inmunología, Instituto de Investigaciones Biomédicas, Universidad Nacional Autónoma de México, AP, Ciudad de México, México

ᴼ These authors contributed equally to this work.
† Deceased.
‡ These authors also contributed equally to this work.
* jmontor66@biomedicas.unam.mx

**Data Availability Statement:** All relevant data are within the paper.

**Funding:** This work was supported by IN-209719 Programa de Apoyo a Proyectos de Innovación

## Abstract

The impact of performing exercise on the immune system presents contrasting effects on health when performed at different intensities. In addition, the consequences of performing chronic exercise have not been sufficiently studied in contrast to the effects of acute bouts of exercise. The porpoise of this work was to determine the effect that a popular exercise regimen (chronic/moderate/aerobic exercise) has on the proportion of different immune cell subsets, their function and if it affects the cannabinoid system with potentially functional implications on the immune system. A marked increase in several immune cell subsets and their expression of cannabinoid receptors was expected, as well as an enhanced proliferative and cytotoxic activity by total splenocytes in exercised animals. For this study male Wistar rats performed treadmill running 5 times a week for a period of 10 weeks, at moderate intensity. Our results showed a significant decrease in lymphocyte subpopulations (CD4+, Tγδ, and CD45 RA+ cells) and an increase in the cannabinoid receptors expression in those same cell. Although functional assays did not reveal any variation in total immunoglobulin production or NK cells cytotoxic activity, proliferative capability of total splenocytes increased in trained rats. Our results further support the notion that exercise affects the immunological system and extends the description of underlying mechanisms mediating such effects. Altogether, our results contribute to the understanding of the benefits of exercise on the practitioner´s general health.

Tecnológica (PAPIIT), Dirección General de Asuntos del Personal Académico (DGAPA), Universidad Nacional Autónoma de México (UNAM) to JMM; Grant FC-2016-2125 from Fronteras en la Ciencia, Consejo Nacional de Ciencia y Tecnología (CONACYT) Programa de Apoyo a Proyectos de Innovación Tecnológica (PAPIIT), Dirección General de Asuntos del Personal Académico (DGAPA), Universidad Nacional Autónoma de México (UNAM) to JMM; Grant IA202919 Programa de Apoyo a Proyectos de Innovación Tecnológica (PAPIIT), Dirección General de Asuntos del Personal Académico (DGAPA), Universidad Nacional Autónoma de México (UNAM) to KENC. The funders had no role in study design, data collection and analysis, decision to publish, or preparation of the manuscript.

**Competing interests:** The authors have declared that no competing interests exist.

## Introduction

The beneficial impact of exercise on the practitioner's general health is a well-known fact [1–3]. Studies suggesting a connection between physical activity and the improvement of health have generated information that ranges from describing adipose tissue loss to changes in genetic expression [4,5]. It is well known that the proper composition of the immune system (IS) and its correct function will allow it to actively overcome challenges that otherwise would compromise the organism's health, such as infections, autoimmune diseases, cancer, etc. It is worth mentioning that the acknowledgment of those benefits conferred by the regular practice of exercise has led to its implementation as an alternative/complementary therapy against metabolic diseases [6,7], and in recent works even for the treatment of cancer [8].

Nonetheless, data regarding the changes induced by exercise in cell subpopulations of the IS and their function seems to be controversial [9–11]. This may be partly explained by the use of different exercise paradigms. Along with this, a classic idea has been established about such changes promoted by exercise, being those models of moderate/chronic exercise the promoters of an enhanced IS, while acute/high intensity models the responsible of hindering it. Finally, evidence suggesting a beneficial effect of the performance of exercise, on modalities previously described as detrimental, has been gathered and old data reinterpreted as in Campbell and Turner's review [12], so nowadays evidence debunking the previous conception of high intensity/long duration or chronic exercise evoked immune suppression increases, at least for several aspects of the immune function. On the same line, many studies document the immediate changes on the IS induced by a single bout of exercise, as opposed to the effects of its chronic performance, which suggest different outcomes and in some cases opposite effects over the IS [11,13,14]. Less attention has been paid to such studies and to the long-term alterations that it may produce on the IS. For instance, macrophages extracted from mice trained for 12 weeks exhibited increased phagocytic activity, superoxide anion production and glucose consumption when compared to macrophages obtained from sedentary mice [11]. Consistently other studies have shown that chronic exercise alters the function of T cells, affecting their production of pro- and anti-inflammatory cytokines, including the up-regulation of IL-2, an important cytokine related to proliferation and activation [14]. Likewise, trained rats presented increased glucose consumption, IL-2 production and IL-2R expression by their lymphocytic subpopulations. Furthermore, those changes obtained in trained animals seem to last days after the last exercise bout [14].

Many molecular pathways that are affected by exercise possess an immunoregulatory potential, ranging from variations in the energy substrates [15,16] to the activation of signaling pathways with direct immune-regulatory relevance, such as: the release of IL 6 by skeletal muscle [4,13], release of stress hormones, catecholamines [4,17] and neurotransmitters by the sympathetic and para-sympathetic system, among others. Hence, in order to contribute to the further understanding of these effects we decided to evaluate the cannabinergic system (CBS). Regarding this system, some studies have reported a subtle increase of anandamide, a widely studied molecule that acts as a CB1 and CB2 receptor agonist, after short bouts of aerobic exercise. Such increase was sustained up to several minutes after the conclusion of the physical activity [18–20]. Furthermore, both receptors are widely distributed in the immune cells and IS structures [21–23] and when activated, together or independently, produce changes in the function of several immune cells, suggesting that their activation is able to modulate the IS function. These modulatory actions have been explored *in vitro* [24,25] and *in vivo* [23]. Likewise, the expression of cannabinergic receptors (CBR) on the surface of immune cells, varies according to their activation and inflammatory status. Given that new data suggests its relevance as an immune-modulatory system, the expression of these receptors provides us with interesting and relevant information about the IS status.

Altogether, the objective of our study is to explore the long-term changes that chronic exercise (CE) produces in the proportion of splenocytes from the adaptive and innate immunity, and to assess the effects that it has on their function (by performing proliferation tests and cytotoxicity test with total splenocytes *in vitro*), and finally to determine if the expression of CBR in these cells is affected by this model of exercise. Furthermore, coherently with previous works on this topic, we expect that the changes evoked by CE will alter the composition of several of the immune cell subsets studied, mainly those from lymphoid origin. CE will also boost their cytotoxic activity and proliferative capacity. In terms of the cannabinoid receptor expression, we anticipate that CE will increase their expression in the entire cell subsets studied.

Finally, this study provide relevant information about those changes elicited on the IS by a way of exercising chosen by a big proportion of the society nowadays. Besides, our results further support the notion that exercise affects the IS and extends the description of underlying mechanisms mediating such effects.

## Materials and methods

### Ethic statement

Animal care and experimental practices were conducted at the Animal Facilities of the Instituto de Fisiología Celular (IFC), Universidad Nacional Autónoma de México (UNAM). All procedures in the experimental animals were approved by the Institutional Care and Animal Use Committee (CICUAL), adhering to Mexican regulation (NOM-062-ZOO-1999), in accordance with the recommendations from the National Institute of Health (NIH) of the United States of America (Guide for the Care and Use of Laboratory Animals). Euthanasia of experimental animals was performed in a humanitarian way.

### Animals

For this study, male Wistar rats ranging between 250 to 300 g were used, proceeding from our own breeding at the animal facilities of IFC, UNAM. The animals were housed at IFC with controlled temperature (22˚C) and 12h light-dark cycles, with water and Purina LabDiet 5015 chow *ad libitum* (Purina, St. Louis MO). Sacrifice of those animals used exclusively for flow cytometry or samples extraction for cellular culture, was carried out by cervical dislocation after pentobarbital sodium (Pisabental®, México) anaesthesia. Animals that also were used for the extraction of the brain were sacrificed by an overdose of pentobarbital sodium (Purina, St. Louis MO). All procedures were carried out in a humanitarian way ensuring the maximisation of efforts in order to alleviate suffering.

### Exercise protocol

Animals were set in one of three experimental groups: Exercised (EXE), Treadmill control (TC) and sedentary group (SED). Animals in the exercised group performed treadmill running 5 times a week for a period of 10 weeks, for which a previous habituation of one week was completed. During the habituation week, animals were placed inside the treadmill and then it was turned on at minimum capacity (4m/min) for 5 minutes per day. Once the habituation period was completed, animals started training. On the first day of training, rats ran at 7.5 m/min for 10 minutes, then speed and duration of exercise was escalated gradually each consecutive day, in order to achieve a daily exercise bout of 40 minutes at 15 m/min by the third week. Remaining weeks of training were kept constant in speed and duration until the sacrifice of the animals. It is noteworthy, that this training protocol equals to moderate chronic exercise, as it has been demonstrated in the works by Pilis and Carvalho [26,27] through the calculation

of the velocity at the lactate threshold. No electrical stimulus was used to incentive animals to run inside the treadmill.

Animals sited in the TC group were placed inside the treadmill at minimum capacity (4m/min) for 10 minutes, 5 times per week, for the same period of time than the exercised group (10 weeks). While being inside the treadmill, animals from TC group were exposed to the same context than animals from the EXE group without being exercised, reflecting any effect in the results prompted by sources other than exercise itself. Animals conforming the SED group were kept alive in standard conditions for the same amount of time than the other two groups.

At the end of the training period animals from every group were allowed to rest for one day in order to eliminate any possible effect of acute exercising. Afterwards, animals were sacrificed either by an overdose of pentobarbital sodium or by anesthesia with the same product followed by cervical dislocation (Pisabental®, México), afterwards the samples were taken.

## Flow cytometry

Spleens were manually disaggregated using a 50µm nylon mesh, and cells resuspended in PBS. Erythrocytes in the solution were lysed using ACK buffer (150 mM $NH_4Cl$, 10mM $KHCO_3$, 0.1mM $Na_2EDTA$, pH 7.3) for 10 minutes and washed three times with PBS, then cells were resuspended in FACS buffer (PBS, FBS, 0.02% $NaN_3$).

Approximately $1x10^6$ cells were incubated with the following antibodies in order to characterize spleen immune subpopulations: Alexa Fluor® 488-conjugated- anti rat CD3 (IF4, Biolegend), PE Cy5-conjugated- anti rat CD4 (biolegend), PE-conjugated anti rat CD8a (Biolegend), PE-conjugated -anti rat CD45RA (Biolegend), Alexa Fluor® 647-conjugated- anti rat CD161 (biolegend), biotin-conjugated- anti rat CD11b (OX-42, Bioloegend), PE-conjugated anti rat TCR (V65, Biolengend). For the determination of macrophages, additionally to the use of anti CD11b antibody, a different gating was used, corresponding to bigger and more complex cells, in order to exclude other cells from the myeloid linage. Staining of the cell subpopulations was made by duplicate in order to additionally mark on each set one of the cannabinoid receptors.

For the staining of cannabinoid receptors, the polyclonal primary antibodies used were: rabbit anti Cannabinoid receptor I (abcam®) and rabbit anti Cannabinoid receptor II (abcam®), followed by the Secondary antibodies: AlexaFluor® 488- conjugated goat anti rabbit IgG (ThermoFisher Scientific) and DyLight® 649- conjugated- anti rabbit IgG (Vector laboratories).

In order to assess proliferation, a Cell Trace™ CFSE cell proliferation kit was used (Invitrogen™). Propidium iodide was used to determine viability of cells during cytotoxic assays. Attune Cytometer (life technologies) was used to obtain data which was further analyzed with FlowJo software (Treestar Inc.).

## Proliferation assays

Total splenocytes were obtained as previously described and quantified on a Neubauer chamber. Subsequently, splenocytes were marked with CFSE cell tracer, which was used according to the manufacturer's protocol. Finally, cells were resuspended in RPMI-1640 medium (ATCC® 30 2001™). Splenocytes were cultivated in RPMI-1640 medium plus Ionomicyn (SIGMA-ALDRICH® 10634™) and PMA (SIGMA-ALDRICH® P8139-1MG™) at concentrations100 nm and and 25ng/ml for 72 hours. Proliferation was assessed using an Attune cytometer (Life Technologies) with blue and red lasers, obtained data was further analyzed with FlowJo software (Treestar Inc.).

## Cell culture

Yac 1 cell line (ATCC® - TIB 160™) was cultivated in RPMI 1640 medium (ATCC® 30–2001™) supplemented with 10% Fetal bovine serum (FBS, ATCC® 30–2020™) and kept with air, 95%; carbon dioxide ($CO^2$), 5% at 37°C. Yac 1 cells were expanded for a week then counted on a Neubauer chamber and finally stained with CFSE kit for further use in cytotoxic assays.

## *In vitro* cytotoxic assay

Yac 1 cells were stained with cell trace™ CFSE kit according to the manufacturer's protocol and used as target cells for the assay. A single cell splenocyte suspension was obtained from rats in the different experimental conditions as previously described, they were counted and used as the effector cells in the assay. Finally both, effector and target cells were co-cultivated in RPMI medium supplemented with 10% FBS for 4 hours, into 96 round well plates. Thereafter co-cultures were removed from the incubator and stained with Propidium Iodide and washed with Facs buffer. Acquisition was performed in an Attune Cytometer (life technologies) and data further analyzed with FlowJo software (Treestar Inc.).

## Corticosterone and IgG levels assesssment

Animals were anesthetized and sacrificed one day after concluding their experimental condition, at the same time that they had been set for exercising (14:00–16:00 hr.). Cardiac puncture was performed in order to extract blood, which was immediately centrifuged at 4000rpm to collect the blood serum. Subsequently, blood serum was divided into two aliquots and stored at -70°C for later use. Corticosterone levels were assessed with a Corticosterone ELISA kit (Abcam® ab-108821) and procedures underwent according to manufacturer's protocol. Assessment of the levels of IgG in serum were carried out on a 96 flat bottom well plate. The plate was previously sensitized with a dilution of blood serum (1:1000), washed and blocked with a 1% albumin solution. Subsequently, the plate was incubated for 2 hours antibodies α IgG rat HRP were incubated, and once incubation finished, several washes were performed, chromogen was added to the wells, the reaction was stopped and the reading of the plate was carried out on a Stat Fax 4200 microplate reader (Awarness Technology).

## Statistical analysis

For data regarding the changes of every cell subpopulation, a one-way ANOVA ($\alpha = 0.05$) was performed followed by a Tukey post-hoc test. Differences were considered significant when $p < 0.05$, with the actual p value and n being stated in each figure legend. Before the selection of the ANOVA test the normal distribution of the data was assessed via Shapiro-Wilk test. A similar process was carried out for the statistical analysis of the data regarding the proliferation and cytotoxicity tests, as well as for the data concerning the levels of corticosterone and IgG in serum. For the assessment of the expression of CBR, a two-way ANOVA ($\alpha = 0.05$) was performed, because of the consideration of two independent variables (group and CBR), followed by a Bonferroni post-hoc test with the same significant difference criterion. Data from all the experiments were charted as mean ± standard error, and analysed with Prism 5 software for Mac (GraphPad Software Inc.)

# Results

## Chronic-moderate exercise affects body composition parameters but not total ingestion of food or water

After ten weeks of rats being exposed to each experimental condition, animals were weighted, and results showed a significant gain of weight on those animals corresponding to both control

groups (SED, 490.8g; TC, 495.8g) against those from the EXE group (452g). ANOVA, p = 0.0007, n = 6, P<0.05. Accordingly, when samples were extracted, an observable difference in fat accumulation was perceived among the experimental groups, being those animals that performed the physical activity the ones with the lower amount of visceral fat. Interestingly water and food ingestion per week did not change among animals from the different experimental groups (p = 0.3309, n = 6, P<0.05, and p = 0.3312, n = 6, P<0.05 respectively).

## Chronic-moderate exercise alters the composition of splenocyte subpopulations

Distribution of immune cells is a parameter that provides information about deficiencies or alterations from the IS, therefore we decided to evaluate several cell subpopulations from the innate and adaptive immune system in the spleen of rats that underwent different experimental conditions (S1 Fig). The immune cell subpopulations from the innate immune response that were analyzed corresponded to: NK cells (CD161+) and macrophages (CD11b+) for which flow cytometry analysis did not reflect a significant difference among groups (Fig 1). Analyzed cells from the adaptive immune response were: total T lymphocytes (CD3+), T helper lymphocytes (CD4+), cytotoxic T lymphocytes, Tγδ lymphocytes (Tγδ+) and B lymphocytes (CD45 RA+). For those cells studied, flow cytometry analysis reflected a decrease in the proportion of T helper lymphocytes (Fig 2) and in B lymphocytes (Fig 3) from the EXE group when compared to both control groups, SED and TC (Fig 2 and Fig 3), therefore considering such changes an effect of CE. On the other hand Tγδ lymphocytes showed an increase in the EXE group when compared to SED and TC groups (Fig 2), once more reflecting a change attributable to training. On the other hand, T cytotoxic cells decreased in TC and EXE groups in contrast to SED control group, reflecting an effect non attributable to exercise, but to the exposure to the treadmill (Fig 2). Total T lymphocytes (CD3+), did not show changes among the experimental groups (Fig 2).

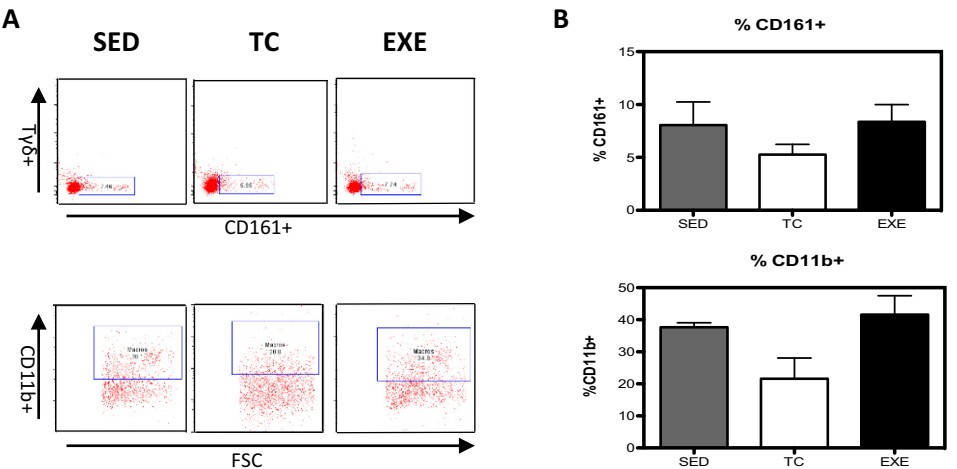

**Fig 1. Changes in the composition of splenocyte populations related to the innate immune response.** (A) Representative dot plot of the cytometric analysis of the subpopulation percentages. (B) Determination of splenocyte subpopulations from the innate immune response one day after being exposed to each condition in the different groups: SED, TC and EXE; data from 4 independent experiments are expressed as mean ± SE. No subpopulation showed statistically significant changes: Natural killer cells (ANOVA, p = 0.0683, n = 10), and Macrophages (ANOVA, p = 0.0273, n = 10).

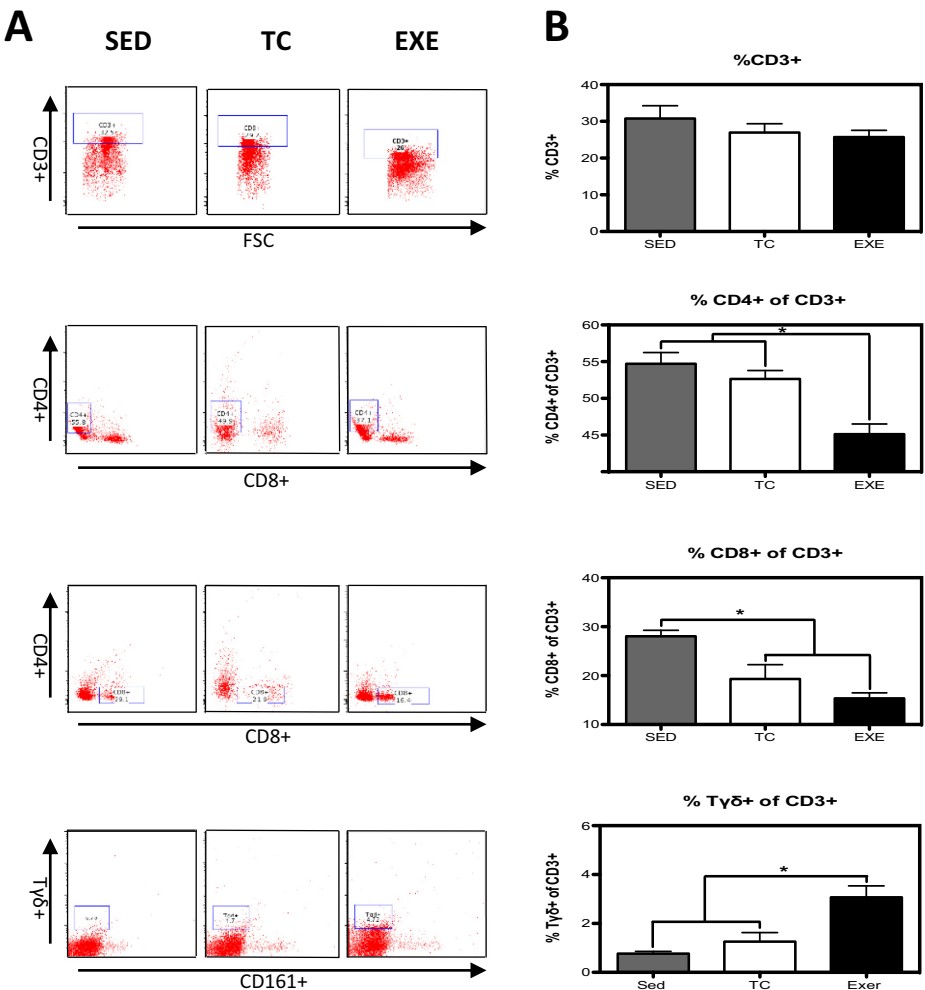

**Fig 2. Changes in the composition of splenocyte populations related to the adaptive immune response.** (A) Representative dot plot of the cytometric analysis of the subpopulation percentages. (B) Determination of splenocyte populations from the adaptive immune response in the different groups: SED, TC and EXE; data from 4 independent experiments are expressed as mean ± SE. Subpopulation that showed statistically significant changes were: T helper lymphocytes (ANOVA, p = 0.0008, n = 5) and Tγδ lymphocytes (ANOVA, p = 0.0002, n = 10). T lymphocytes (ANOVA, p = 0.3739, n = 12), cytotoxic T lymphocytes (ANOVA, p = 0.0016, n = 12). * Means statistically different from the two other groups, P<0.05.

## Modulation of CBR Expression in splenocytes after chronic exercise

CBR are widely distributed among immune cell subpopulations and structures from the IS. The expression of CBR on immune cells has been demonstrated to vary depending on activation or inflammatory profile, among other parameters. Thus, we decided to evaluate if CE would promote changes in the expression of CBR on splenocytes.

From the innate immune system, NK cells (CD161+) and macrophages were analyzed. NK cells from EXE and TC groups showed a decrease in CB2 expression, compared to SED, while no change was observed in CB1 expression. Macrophages (CD11b+) did not present changes in the expression of any CBR among groups (Fig 4). From the adaptive immune response, T helper lymphocytes (CD4+ cells) presented an increase in the expression of CB1 in animals from EXE group when compared to both control groups SED and TC, while no statistically

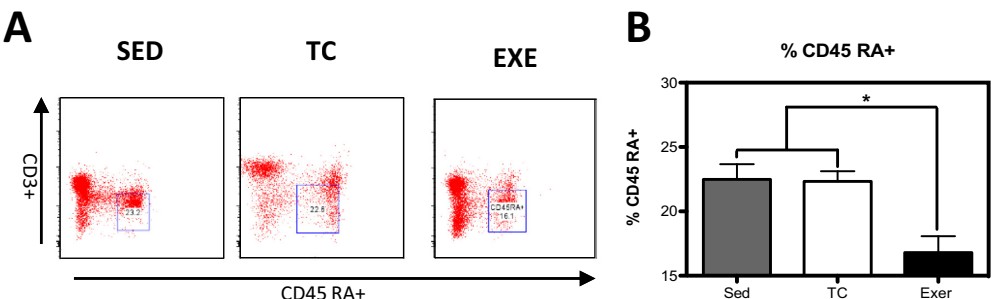

**Fig 3. Changes in the composition of B-lymphocytes from spleen.** (A) Representative dot plot of the cytometric analysis of the subpopulation percentages. (B) Determination of B-lymphocytes of spleen of rats in the different groups: SED, TC and EXE; data from 4 independent experiments are expressed as mean ± SE. B lymphocytes showed a statistically significant change (ANOVA, p = 0.3601, n = 10). * Means statistically different from the two other groups, P<0.05.

significant difference was observed regarding CB2 expression. A similar phenomenon was observed in Tγδ subpopulation from EXE animals, which showed a higher expression of CB1 when compared to SED and TC groups (Fig 5), with no change in the expression of CB2. The expression of CBR did not vary in the subpopulations of T lymphocytes (CD3+) and cytotoxic T lymphocytes when experimental groups were compared (Fig 5).

On the other hand, B lymphocytes from the spleen of EXE animals showed a significant increase in CB2 expression, compared to those from SED and TC groups, whilst no change was reflected between groups in the expression of CB1 (Fig 6).

## Immunoglobulin G levels are not altered by chronic exercising

Immunoglobulin G (IgG) is the most abundant type of immunoglobulins and a reliable parameter to assess the function of plasmatic cells. A change in the amount of IgG could represent an ongoing infectious process or an alteration on the normal function of plasmatic cells when observed in intact animals. We decided to assess if the total production of IgG would vary among our experimental groups. To do so, we performed a direct semi-quantitative ELISA. When data was analyzed statistically (ANOVA, n = 6, p = 0.0676) results did not show

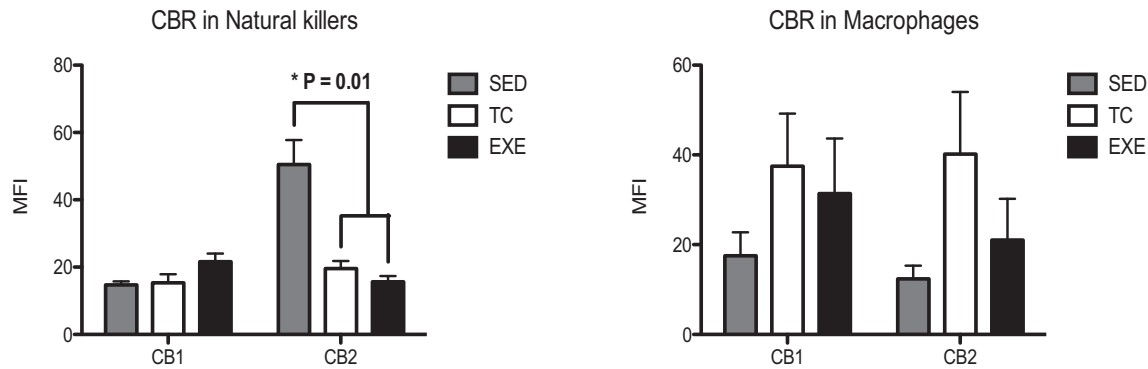

**Fig 4. Expression of CBR in splenocyte populations related to the innate immune response.** Analysis of the expression of CB1 and CB2 CBR in splenocyte populations from the innate immune response (NK's and macrophages) among experimental groups: SED (shaded bar), TC (white bar) and EXE (Solid bar); data from 4 independent experiments are expressed as mean ± SE. Lines connecting bars represent comparison among groups, * p<0.05. Two way ANOVA and Bonferroni post-test, n = 9.

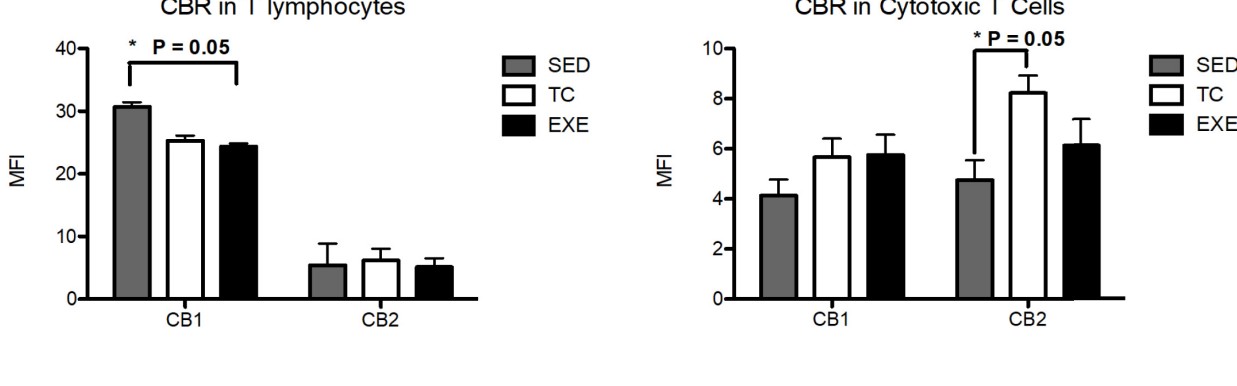

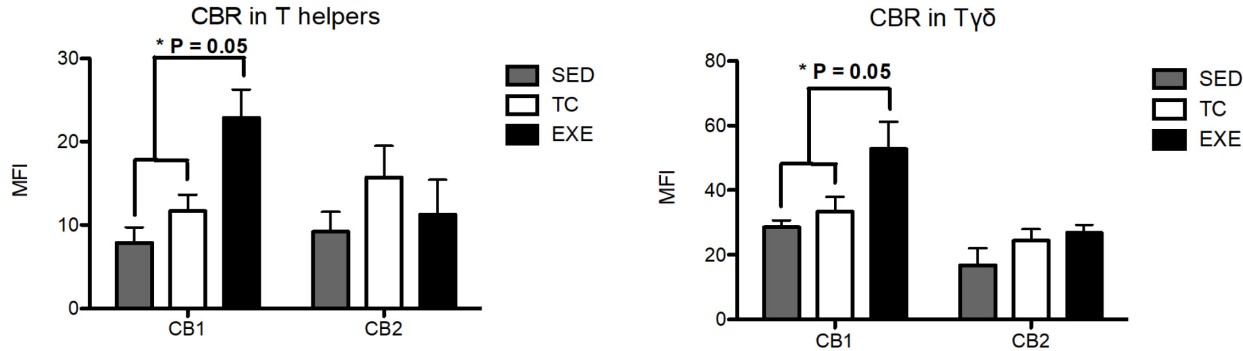

**Fig 5. Expression of CBR in splenocyte populations related to the adaptive immune response.** Analysis of the expression of CB1 and CB2 CBR in splenocyte populations from the adaptive immune response (T lymphocytes; n = 10, T helper lymphocytes; n = 8, cytotoxic T Lymphocytes; n = 10 and Tγδ n = 8) among experimental groups: SED (shaded bar), TC (white bar) and EXE (Solid bar); data from 4 independent experiments are expressed as mean ± SE. Lines connecting bars represent comparison among groups, * p<0.05. Two way ANOVA and Bonferroni post-test.

any significant difference among the experimental groups: SED (optic density 2.731),TC and EXE (OD 2.973 and 2.871 respectively, S2 Fig).

## Chronic exercise enhances proliferative capacity but not cytotoxic activity of total splenocytes

Proliferative capacity and cytotoxic activity have been tested before in order to assess the degree of competence of a subject's immune system. In order to observe if chronic exercise has any effect on both features we performed *in vitro* tests. For the proliferation assay we obtained total splenocytes and cultivated them on complete RPMI medium plus PMA and ionomycin for 72 hours. Data analyzed reflected no difference among groups in their proliferation index (Fig 7B). Nevertheless, total splenocytes from EXE group showed a higher proportion of dividing cells when compared to SED and TC groups (Fig 7C). Furthermore, we decided to test the proliferative capacity of the immune cell subpopulations and we found that B lymphocytes and NK cells from the EXE group exhibited a higher proportion of dividing cells than both SED and TC groups (Fig 7D).

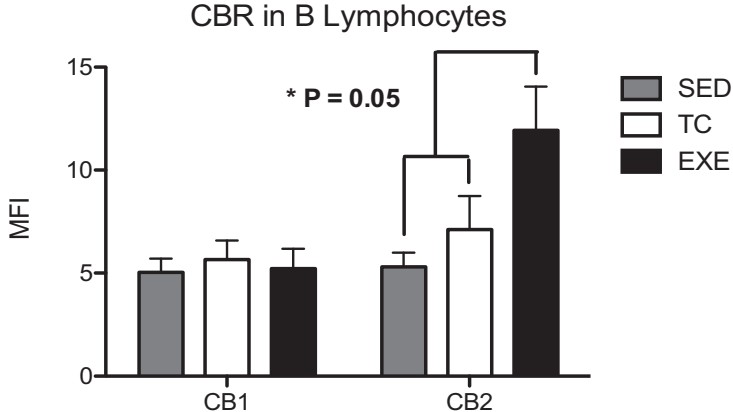

**Fig 6. Expression of CBR in B lymphocytes from spleen.** Analysis of the expression of CB1 and CB2 CBR in B lymphocytes from spleen among experimental groups: SED (shaded bar), TC (white bar) and EXE (Solid bar); data from 4 independent experiments are expressed as mean ± SE. Lines connecting bars represent comparison among groups, * p<0.05. Two way ANOVA and Bonferroni post-test, n = 10.

In order to measure the cytotoxic capacity of splenocytes from the different groups, we performed a cytotoxicity test utilizing Yac1 cells as target cells and cultivating them with total splenocytes from animals of the different groups. Yac1 cells were previously marked with CFSE, and at the end of the test dead cells were dyed with propidium iodide, so double positive cells represented the target cells killed. The assay was carried out at three different effector/target ratios: 10:1, 50:1 and 100:1 and none showed any significant difference among treatments after the statistical analysis (Fig 8).

## Corticosterone level is not altered by chronic exercise

Corticosterone level is considered a reliable stress marker in animals. In order to further comprehend the data obtained, we decided to analyze if corticosterone levels of blood serum differ among groups, reflecting a possible long-term effect of stress in exercised animals. When data was statistically analyzed (ANOVA, n = 6, p = 0.0473, Tukey's) results show a significant difference among the experimental groups. Nonetheless, The post hoc analysis did not show any significant interaction among the experimental groups. Means of the groups: TC and EXE (102.6 and 96.5 ng/ml respectively) were notoriously higher than that of the SED group (46.8 ng/ml, S3 Fig).

## Discussion

Although widely studied, the consequences of physical activity over the IS remain as a promising field not only to expand our comprehension on basic physiology, but also on physiopathology and the different processes that take part on the orchestration of the immune response. At first instance, our investigation tries to emulate in rats a popular paradigm of exercising in modern times across the global population, which involves the chronic performance of medium intensity aerobic exercise. Such particularities in our model have led us to asses its raw impact over the IS composition and its function. Notwithstanding, the vast majority of studies have focused in the effects of short bouts of exercise over the immediate changes in composition and function of the IS [1,11,28,29], and neglecting those focused on the effect of CE and its long lasting effects over the IS [13,30]. In this study we observed that components

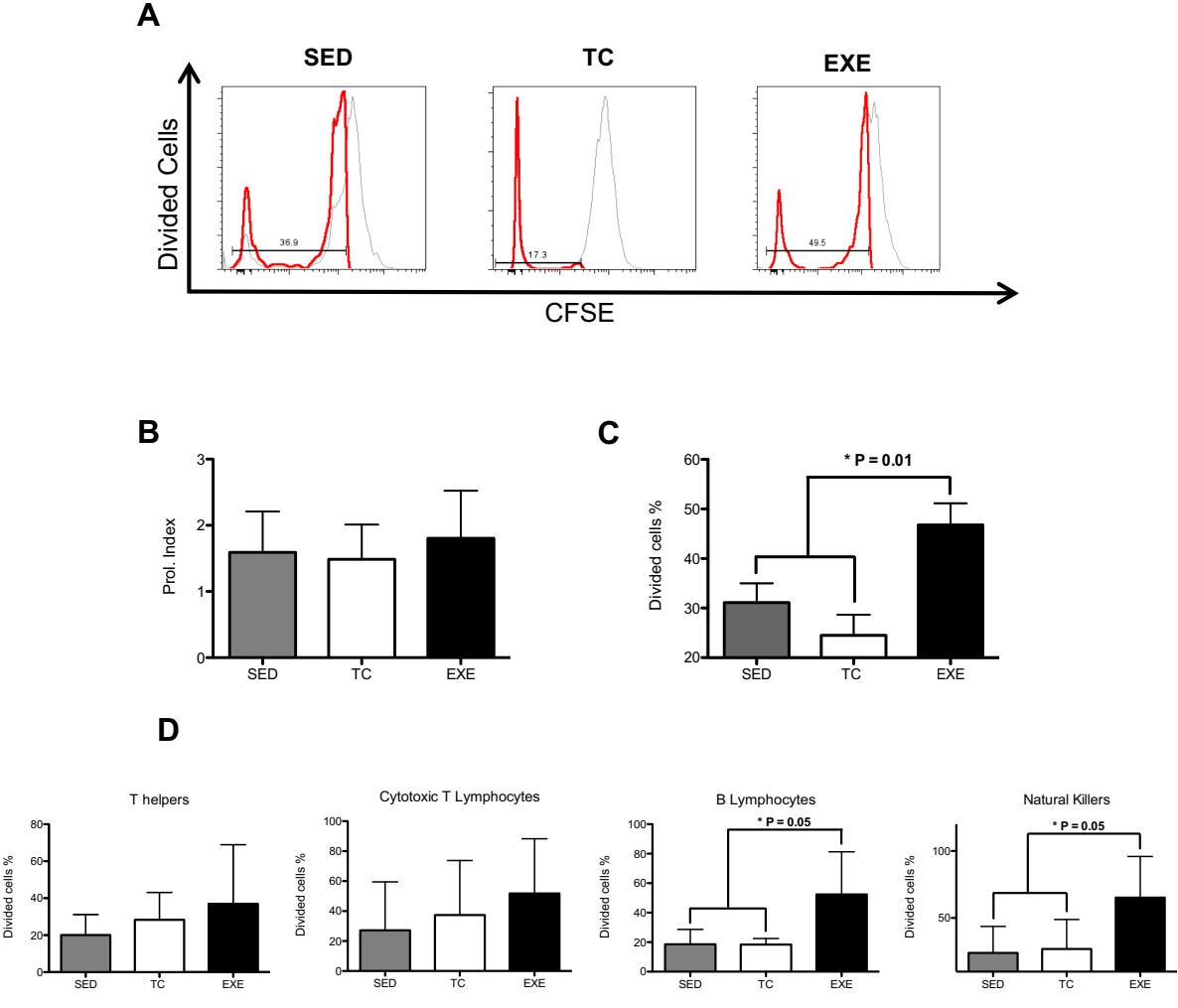

**Fig 7. Effect of chronic exercise on splenocyte proliferation.** (A) Representative histograms of cytometric analysis of dividing cells. Two parameters were considered to assess the proliferative capacity of splenocytes: (B) the proliferation index, which did not change due to chronic exercise and (C) the proportion of dividing cells, where we observed an increase in the exercised group when compared to both control groups (ANOVA, p = 0.0092, n = 5). (D) Analysis of proliferative capacity of specific splenocyte subpopulations: T helper lymphocytes (ANOVA, p = 0.4543, n = 5) and cytotoxic T lymphocytes (ANOVA, p = 0.5248, n = 6) did not show any change between groups; B lymphocytes (ANOVA, p = 0.0006, n = 6) and Natural killers (ANOVA, p = 0.0191, n = 6) from EXE did show an in increase in the proportion of dividing cells when compared to both control groups. In graphic bars SED is represented by shaded bar, TC by white bar and EXE by the solid bar.

from the innate immune response were not affected by chronic-moderate exercise, while elements from the adaptive immune response did change in those animals that underwent physical training. T helper lymphocytes and B lymphocytes were decreased in trained animals, contrary to what would be expected according to the popular statement of moderate exercise enhancing a pro-inflammatory state [31,32]. On the other hand, Tγδ lymphocytes increased in animals from the EXE group, increase that could reflect an intensification in surveillance and protection of the upper respiratory tracts (URT) and mucosa tissue, idea that would be in accordance with the strengthened resistance against URT infections due to moderate exercising and opposite to the supposed effect of higher susceptibility to these infections in high performance athletes [29,33–37]. Nonetheless, the reduction of T helper and B lymphocyte populations was unexpected, since these cells play a major role at recognizing antigens and

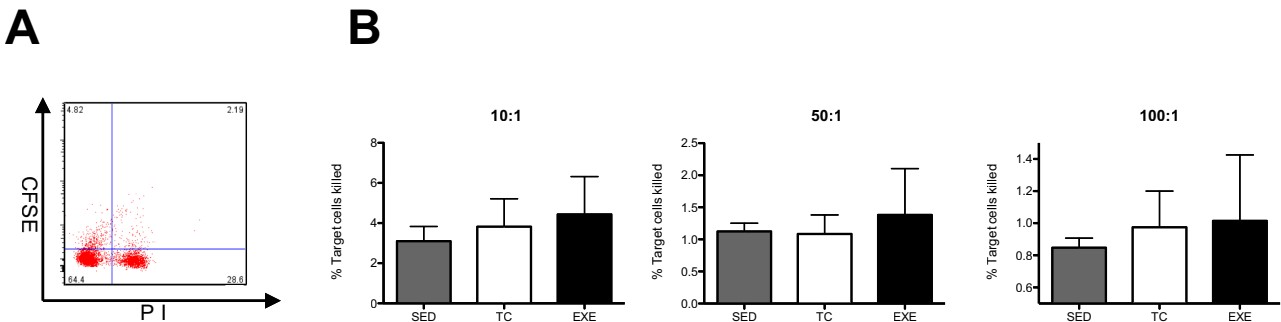

**Fig 8. Cytotoxic activity of total splenocytes in vitro.** (A) Representative dot plot of cytometric analysis of killed target cells percentage. (B) Cytotoxic activity of total splenocytes was assessed in vitro at three different ratios (effector cells: target cells, 10:1, 50:1 and 100:1); data from 2 independent experiments are expressed as mean ± SE, with an n = 6 for each condition. No differences were found among the groups at any of the different ratios. In bar graphics SED is represented by the shaded bar, TC by the white bar and EXE by the solid bar.

therefore at orchestrating immune responses against new and already known threats and once activated they can polarize towards a pro-inflammatory or anti inflammatory state which modulates the activity of several other components from the innate and adaptive immune response. Consequently, the decrease of T helper lymphocytes as well as that of B lymphocytes made us wonder if it could be translated into a deficient immune response of animals of the EXE group.

Once we determined that our exercising paradigm did affect the composition of immune cells subpopulations, our next goal became to assess if those changes would be translated into functional alterations. To accomplish our purpose, we determined total IgG in serum among experimental groups and we also performed proliferation and cytotoxicity test *in vitro*. On the proliferation test, two parameters were evaluated: the proliferation index and the percentage of dividing cells. The first represents the mean of divisions that dividing cells underwent during the assay, which showed no difference among groups. In turn, the percentage of dividing cells among groups during the experiment showed statistical differences, being higher in splenocytes from the EXE group. Altogether, this data suggests that cells from EXE group are not more efficient at dividing once they have been activated but that more cells in proportion from EXE group are prone to proliferate once they have been exposed to PMA and ionomycine. These results made us wonder if this trend would be persistent in basal conditions, so we compared the percentage of proliferation in splenocytes among the experimental groups without activation by PMA and ionomycine and the statistical analysis showed no difference among them, reflecting a response produced by activation and not an anomaly that could reflect an inflammatory state that in turn could favor an autoimmune response. Thereafter, we evaluated the percentage of dividing cells from specific subpopulations, T helper cells, cytotoxic T cells, B lymphocytes and NK cells, showing an increase in the percentage of dividing cells in the last two subpopulations by effect of exercise. On the other hand, statistical analysis from the cytotoxicity test did not show variation among the experimental groups in any of the target/effector cell ratios tested, so data provided from this experiment does not suggest a higher cytotoxic activity from NK cells as a consequence from exercise. Nonetheless, exercise enhances the amount of NK cells that proliferate, which may indicate a higher immune-surveillance against transformed and virally infected cells in chronically exercised subjects.

Even though we found a decrease in major lymphocyte subpopulations (CD4+ and CD45 RA+ cells), which has been reported before for other immune cells in long term exercised individuals [32,38–40], we also determined that a bigger proportion of splenocytes is prone to

activate when stimulated, plus other functions of splenocytes from the EXE animals were not impaired. Given the fact that long-term exercised subjects do not report any kind of immune suppression, we have come to hypothesize that the decrease in the composition of some cell subpopulations may represent a more efficient IS, which requires less elements but presents a stronger reaction when needed. This idea is also supported by the results shown by the ELISA test, which showed that levels of IgG did not change among the groups, even though the sub-population of B lymphocytes is decreased in EXE animals.

Finally, recent works have denominated the endocannabinoid system (ECS) as an immuno-modulatory system; inhibiting the function of highly reactive and pro-inflammatory cells [25,41,42]. The ECS exerts its functions through the activation of its receptors, which expression vary greatly depending on cell subpopulation, activation or inflammatory status [24,41,43,44], being increased in the surface of more reactive cells. In this work we demon-strated that CE promotes changes in the expression of both CBR's in splenocytes, through the staining of each receptor on different cell suspension aliquots and due to the shared epitope for the secondary antibody. Our findings concur with previous works showing that during moderate exercise bouts there is an increase of circulating ECS agonists [18,19]. We must emphasize that changes in CBR's expression remained after one skipped day of training, which suggests that subjects exercising on a daily basis or in intervals of every two days might be maintaining this alterations. Our methodology allowed us to determine differences in the expression of CBR among cell subpopulations even between control groups. These differences might be explained by the intrinsic variability among cell subpopulations [21,45]. We also assessed an increased expression of CB1 receptor in the subpopulations of T helper and Tγδ lymphocytes and an increase in the expression of CB2 receptor in B lymphocytes of exercised animals. Therefore enhanced expression of CB1 in T helper, Tγδ, and CB2 in B lymphocytes could represent a mechanism to down regulate the activity of these probable high metabolic cells from EXE animals, as reflected by the proliferation tests. Some of these data differ from anterior reports concerning expression of CBR, nonetheless most of those reports used differ-ent techniques and did not measure the protein conforming CBR's but mostly mRNA [21,46,47], which highlights the novelty of the data presented in this work, by showing a direct measure of the expressed protein in question under this particularly conditions of exercise. On the same line, the increased expression of CBR in splenocytes, particularly in lymphocytes from the EXE group, concurs with previous data reporting changes in the expression of the protein through the assessment by different techniques, like western blot and immunofluores-cense [21,48,49], adding validity to our results.

We would like to address those changes presented on this work that can not be attributable to CE, like the composition of cytotoxic T cells for which TC and EXE groups differ when compared against SED group, reflecting an effect relying probably on the stress produced by the placement of the animals inside the treadmill, being that, the one thing that those groups had in common. The same explanation seems plausible for the expression of CBR in NK and cytotoxic T cells. Nevertheless analysis of corticosterone did not showed significant difference among the experimental groups, different sensitivity to several molecules has been reported for the wide variety of cells from the IS, leaving the possibility of other molecular interactions that escaped our control and awareness.

## Supporting information

**S1 Fig. Gating strategy for the flow cytometric analysis of rat spleen subpopulations and their expression of CB1 and CB2 receptors.** Single cell suspension was prepared and stained with fluorochrome-conjugated antibodies to separate splenocyte subpopulations and to mark

cannabinoid receptors (CB1 and CB2). Data was analyzed with FlowJo software 8.7 for Mac. Lymphocytes were identified by their scatter properties (FSC-A x SSC-A plot). Splenocyte sub-populations were characterized by surface staining and gated for their quantity assessment. Subsequently each cellular subpopulation was analyzed for their expression of both cannabinoid receptors in their surface.
(TIFF)

**S2 Fig. Levels of IgG are not affected by chronic exercise.** The analysis of total IgG was assessed for every experimental group with the use of a direct semi-quantitative ELISA. Statistical analysis did not show any significant difference among the experimental groups: SED (shaded bar), TC (white bar) and EXE (Solid bar). P>0.05. ANOVA, p = 0.0676, n = 6.
(TIF)

**S3 Fig. Serum corticosterone concentration one day after the last exercising bout.** Data is shown as mean (ng/ml) +- SE for each group. There was no significant difference among group values in concentration of serum corticosterone. Groups analyzed: SED (shaded bar), TC (white bar) and EXE (Solid bar). When data was statistically analyzed (ANOVA, n = 6, p = 0.0473, Tukey's) results show a significant difference among the experimental groups. Nonetheless, The post hoc analysis did not show any significant interaction among the experimental groups. Means of the groups: TC and EXE (102.6 and 96.5 ng/ml respectively) were notoriously higher than that of the SED group (46.8 ng/ml).
(TIF)

# Acknowledgments

We thank M.V.Z Claudia Rivera-Cerecedo and her animal facility staff for assisting in the breeding of experimental animals. We also thank Diana Millán-Aldaco and Marcela Palomero-Rivero for their technical support.

Salvador Valencia-Sánchez is a doctoral student from Programa de Doctorado en Ciencias Biomédicas, Universidad Nacional Autónoma de México (UNAM) and received fellowship 223625 from CONACYT.

# Author Contributions

**Conceptualization:** Salvador Valencia-Sánchez, Karen Elizabeth Nava-Castro, Oscar Prospéro-García, Jorge Morales-Montor, René Drucker-Colín.

**Data curation:** Karen Elizabeth Nava-Castro.

**Formal analysis:** Salvador Valencia-Sánchez, Karen Elizabeth Nava-Castro, Margarita Isabel Palacios-Arreola, Oscar Prospéro-García, Jorge Morales-Montor.

**Funding acquisition:** Karen Elizabeth Nava-Castro, Jorge Morales-Montor, René Drucker-Colín.

**Investigation:** Salvador Valencia-Sánchez, Margarita Isabel Palacios-Arreola, Oscar Prospéro-García, Jorge Morales-Montor.

**Methodology:** Salvador Valencia-Sánchez, Margarita Isabel Palacios-Arreola.

**Project administration:** Jorge Morales-Montor, René Drucker-Colín.

**Resources:** Oscar Prospéro-García, Jorge Morales-Montor.

**Writing – original draft:** Karen Elizabeth Nava-Castro, Margarita Isabel Palacios-Arreola, Oscar Prospéro-García, Jorge Morales-Montor, René Drucker-Colín.

**Writing – review & editing:** Jorge Morales-Montor.

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
