## [Decision Letter · Decision Letter 0]

29 Aug 2019

PONE-D-19-20066

Cellular innate and adaptive immunity are affected by chronic exercise: implication of the cannabinergic system

PLOS ONE

Dear Dr. Morales-Montor,

Thank you for submitting your manuscript to PLOS ONE. After careful consideration, we feel that it has merit but does not fully meet PLOS ONE’s publication criteria as it currently stands. Therefore, we invite you to submit a revised version of the manuscript that addresses the points raised during the review process.

We would appreciate receiving your revised manuscript by Oct 13 2019 11:59PM. To enhance the reproducibility of your results, we recommend that if applicable you deposit your laboratory protocols in protocols.io, where a protocol can be assigned its own identifier (DOI) such that it can be cited independently in the future. For instructions see: http://journals.plos.org/plosone/s/submission-guidelines#loc-laboratory-protocols

We look forward to receiving your revised manuscript.

Kind regards,

Junpeng Wang, Ph.D

Academic Editor

PLOS ONE

Journal Requirements:

2. To comply with PLOS ONE submissions requirements, in your Methods section, please provide additional information on the animal research and ensure you have included details on (1) methods of sacrifice, (2) methods of anesthesia and/or analgesia, and (3) efforts to alleviate suffering.

This work was supported by Grant IN-209719 and Grant IA 202919 , both from Programa de Apoyo a Proyectos de Innovación Tecnológica (PAPIIT), Dirección General de Asuntos del Personal Académico (DGAPA), Universidad Nacional Autónoma de México (UNAM); to JMM and KENC, respectively. Grant FC-2016-2125 from Fronteras en la Ciencia, Consejo Nacional de Ciencia y Tecnología (CONACYT), also to JMM. Margarita I Palacios-Arreola is a Posdoctoral fellowship from DGAPA, UNAM. We also receive funding from Fideicomiso: bases de colaboración “Transplantes al cerebro” to René Drucker Colín. In addition, Salvador Valencia- Sánchez received a complementary scholarship from Fondo de Estudios e Investigaciones Ricardo J. Zevada, to complete the final stages of this study. Salvador Valencia-Sánchez is a doctoral student from Programa de Doctorado en Ciencias Biomédicas, Universidad Nacional Autónoma de México (UNAM) and received fellowship 223625 from CONACYT.

JMM IN-209719 Programa de Apoyo a Proyectos de Innovación Tecnológica (PAPIIT), Dirección General de Asuntos del Personal Académico (DGAPA), Universidad Nacional Autónoma de México (UNAM).

JMM Grant FC-2016-2125 from Fronteras en la Ciencia, Consejo Nacional de Ciencia y Tecnología (CONACYT) Programa de Apoyo a Proyectos de Innovación Tecnológica (PAPIIT), Dirección General de Asuntos del Personal Académico (DGAPA), Universidad Nacional Autónoma de México (UNAM).

KENC IA202919

Reviewers' comments:

Reviewer's Responses to Questions

**Comments to the Author**

1. Is the manuscript technically sound, and do the data support the conclusions?

Reviewer #1: No

Reviewer #2: Partly

2. Has the statistical analysis been performed appropriately and rigorously? 

Reviewer #1: Yes

Reviewer #2: Yes

3. Have the authors made all data underlying the findings in their manuscript fully available?

Reviewer #1: Yes

Reviewer #2: Yes

4. Is the manuscript presented in an intelligible fashion and written in standard English?

Reviewer #1: Yes

Reviewer #2: Yes

5. Review Comments to the Author

Reviewer #1: Comments to the Author:

The authors examined the effect of chronic treadmill exercise on cells of the immune system and cannabinoid receptors in male rats. These results suggested that chronic exercise can be recommended because it has beneficial effects on cellular innate and adaptive immunity. This paper is interesting to a wider audience across disciplines as immune systems and cannabinergic system. The authors suggest that the results contribute to the understanding of the benefits of exercise on the generally healthy person. Although these research approaches are interesting, to suggest the contribution to health there are several points that need to be clarified.

1. As basically data, it has need to show weight change in effect of Citrate Synthase activity (CS activity) of gastrocnemius muscle and/or weight changes about chronic treadmill exercise in rats. Because, in this experiment, it was conducted a chronic exercise at the vivo level, it needs information about training effect is necessary.

2. I would like to recommend that the authors need to observe the chronic exercise changes in cytokine production associated with the immunomodulation and inflammation about the functions of the other organs or tissues, immune cells.

3. For example, what about the effects of CRH by chronic training? In addition, it is necessary to add the data as to whether CRH changes by chronic training. Because the relationship between the changes in the immune system on exercise training is the effects of the relationship between the metabolic system, autonomic nervous system and endocrine system too. In particular, the exercise-induced cannabinoid is also involved in the central nerve, so it is thought that it is difficult to suggest health effects solely from the response of immunocyte and only receptor expression. Although I have no doubt about the quality of the presented work, I recommend revising the manuscript so that the purpose appears more clearly.

4. I would like to recommend that the authors demonstrate to investigate the immunocyte functional effect of chronic exercise by cannabinoid receptor antagonist, virodhamine administration.

Minor comments:

1. P4,L4: “physycal activity” = “physical activity”

Reviewer #2: I have reviewed the Ms entitled "Cellular innate and adaptive immunity are affected by chronic exercise: implication of the cannabinergic system " by Valencia-Sánchez et al.

Authors assessed the effect of chronic exercise on imune cells from innate and adaptative immunity. The study is interesting, however it needs some adjustments.

1- The title of the manuscript is very long. The authors should summarize it.

2- In abstract the authors indicate that the exercise produced 'an alteration in the cannabinoid receptors expression'. This feels could be better specified in increase or decrease.

1- At the end of the introduction the authors describe many about the results. This can be replaced only by the hypothesis and purpose of the study.

2- The introduction makes it unclear whether exercise activates or inhibits the immune system and whether this has any clinical implications. Has this effect of exercise been investigated in any pathology? It would be interesting to describe in the introduction.

3- In methods, it would be interesting to describe the weight of the animals at the end of the tenth week.

4- Was there an electrical stimulus on the treadmill to encourage the animals to run? Usually in these protocols there is an electrical stimulus. In this case, the animals Treadmill control group received the same average stimuli Exercised animals? Can this stimulus influence the immune response?

5- Does the exercise intensity at the end of the tenth week equate to light, moderate or heavy exercise? The type of intensity can directly influence the results and it is important to describe this sentence in the text.

6- Authors should describe exercise protocol in a table.

7- on page 149 delete a comma.

8- On page 173, were the animals sacrificed 1 or 2 days after the end of the exercise?

9- On page 343 the authors describe "medium intensity resistance exercise". Would it be a resistance or aerobic exercise? How do the authors know that indensity is medium?

10- On page 350 delete the word "that" before "underwent".

11- Authors should discuss and investigate clinical studies of the effect on reducing the anti-inflammatory immune response.

12- On page 396 change "endo-cannabinoid" to endocannabinoid.

13- As the authors could explain the increased expression of CB1 receptors in same cells investigated, whereas CB2 receptors are expressed in most immune system?

6. PLOS authors have the option to publish the peer review history of their article (what does this mean?). If published, this will include your full peer review and any attached files.

Reviewer #1: No

Reviewer #2: No

---

## [Author Response · Author response to Decision Letter 0]

23 Sep 2019

Reviewer #1: Comments to the Author:

Q. The authors examined the effect of chronic treadmill exercise on cells of the immune system and cannabinoid receptors in male rats. These results suggested that chronic exercise can be recommended because it has beneficial effects on cellular innate and adaptive immunity. This paper is interesting to a wider audience across disciplines as immune systems and cannabinergic system. The authors suggest that the results contribute to the understanding of the benefits of exercise on the generally healthy person. Although these research approaches are interesting, to suggest the contribution to health there are several points that need to be clarified.

A. Thank you for your comment. We have incorporated all suggestions/comments, were appropiate on the new draft of the manuscript.

Q. As basically data, it has need to show weight change in effect of Citrate Synthase activity (CS activity) of gastrocnemius muscle and/or weight changes about chronic treadmill exercise in rats. Because, in this experiment, it was conducted a chronic exercise at the vivo level, it needs information about training effect is necessary.

A. Thank you for your remark. However, with all due respect, we do not agree with this comment. The weight data it is included, and animals showed a less gain of weight than controls, with no change in food and water intake. Moreover, as the experiments were conducted on an strict protocol, and, under aproval of the Ethics Comitte of our Institution, a new protocol would have to be submitted to asses your concern, that, surely would be denied if the idea it is only to measure this enzyme activity. Unafortunately we do not include to measure this in our original protocol. The training effect that we search for, it is on immune system components, and, in the search of exercise literatura, most of the mapapers reviewed, did not include this enzyme activity like the proof of chronic exercise activity. That it is the main reason not to included it. .Furthermore, the senior author of this manuscript, Dr Drucker, passed away, and, the PhD student in charge of the research (Salvador Valencia), has not grant to perform new experiments. It is not my research line, so, it is practically imposible to perform new experiments. This would be the paper to be published by SV to obtain his PhD. Finally, since the training effect that we search for, it is on immune system components, and, the animals were quite fit (they did not gain weight as the controls), we did not search for proof of exercise at the metabolic level, which by the way, it is a different question. 

Q. I would like to recommend that the authors need to observe the chronic exercise changes in cytokine production associated with the immunomodulation and inflammation about the functions of the other organs or tissues, immune cells

A. I would like to remark, that, cytokines are molecules that indeed give a clue of specific immune cell activatiòn. However, in my experience, cytokines are molecules that are sinthetized and secreted by immune cwells in response to specific immune activation. This is it, in response to an antigenic challenge. Moreover, even if that happens, the cytokine produced most of the times are only in the compartment of the immune system activation (for instance, if a tumor, in the tumor microenvironment, ora n intestinal parasite, in the intestine). As for your suggestion, were would us will have to search for cytokines? In the serum? In the spleen? In the brain? In the muscle? Again, we have many data, in hundreds of animals, in which, cytokines do not change at systemic level, but in the site of the immune confrontation. As a matter of fact, we did both preliminary experiments searching for pro-inflamatory cytokines in serum, and in the spleen, and, since results did not showed any change, we did not pursued it any more. But, since it were only 3 animals, ww cannot include that information, it is incomplete. And, for new experiments, same argument that in the upper comment apllies, it is imposible to perform new experiments. However, we believe that the present communication stands by it self with no need of new experiments.

Q. For example, what about the effects of CRH by chronic training? In addition, it is necessary to add the data as to whether CRH changes by chronic training. Because the relationship between the changes in the immune system on exercise training is the effects of the relationship between the metabolic system, autonomic nervous system and endocrine system too. In particular, the exercise-induced cannabinoid is also involved in the central nerve, so it is thought that it is difficult to suggest health effects solely from the response of immunocyte and only receptor expression. Although I have no doubt about the quality of the presented work, I recommend revising the manuscript so that the purpose appears more clearly.

A. We are very aware of the stress effect. That it is the reason to measure corticosterone. Since cortiosterone secretion it is regulated by the hyphotalamic-pituitary-adrenal (HPA) axis, and, the mechanism of production of hormones is by up or down regulation, and, because we did not found changes in corticosterone level, we did not search for CRH or ACTH.And, I do not follow what do you mean with the purpose appears more cleraly, right after the CRH suggestion. I beleieve, again, that our aim, and way to demostrate our hypothesis it is right, and, there is no need to include or pursue CRH changes. Again, and, most importantly, that it would be only if chnges in corticosterone would have been found, but, as it is no change, there is no need to search for CRH or ACTH changes, because the mechanism of corticosterone production it is by up or down regulation: a decrease in corticosterone, would induce and increase in CRH and ACTH, and an increase, would produce a decrease in CRH and ACTH. If the concern it is the centrl nervous system, the molecules to search are either endrphins or adrenaluine and noradrenaline, that are more related to excercise effects. Those data we are analyzing now, but, may be to expose in another communication. 

Q. I would like to recommend that the authors demonstrate to investigate the immunocyte functional effect of chronic exercise by cannabinoid receptor antagonist, virodhamine administration.

A. That suggestion it is well taken, but, we believe again, that, it it not necessary in the present set of expermients. As much of the scientific communications, not all the mechanisms are explored in the same piece of work. And, unafortunately, I have to remark the comment made in the first of your concerns: we cannot perfomr new experiments. 

Minor comments:

Q. P4,L4: “physycal activity” = “physical activity”

A. It has been corrected as suggested

Reviewer #2: Comments to the Author:

 I have reviewed the Ms entitled "Cellular innate and adaptive immunity are affected by chronic exercise: implication of the cannabinergic system " by Valencia-Sánchez et al. Authors assessed the effect of chronic exercise on imune cells from innate and adaptative immunity. The study is interesting, however it needs some adjustments.

Q- The title of the manuscript is very long. The authors should summarize it.

A. It has been corrected as suggested.

Q- In abstract the authors indicate that the exercise produced 'an alteration in the cannabinoid receptors expression'. This feels could be better specified in increase or decrease.

A. It has been corrected as suggested

Q. At the end of the introduction the authors describe many about the results. This can be replaced only by the hypothesis and purpose of the study.

A- It has been corrected as suggested

Q- The introduction makes it unclear whether exercise activates or inhibits the immune system and whether this has any clinical implications. Has this effect of exercise been investigated in any pathology? It would be interesting to describe in the introduction.

A. Thank you for your comment. Indeed, that it is the whole point of our manuscript. Nothing has been done in regards to chronic exercise, and immune activation, much less in any disease. That it is the reason we do not specifically use manuscripts related to that. 

Q- In methods, it would be interesting to describe the weight of the animals at the end of the tenth week.

A. That information has been included in the result section

Q- Was there an electrical stimulus on the treadmill to encourage the animals to run? Usually in these protocols there is an electrical stimulus. In this case, the animals Treadmill control group received the same average stimuli Exercised animals? Can this stimulus influence the immune response?

A. There is not electrical stimulus. You are absolutely right in your comment. But, in our experiments, rats that were included in the chronic exercise group were those that willingly entered to the treadmill. We also though about describe those as the “bold rats”, compared to the ones that, didi not wanted to get in. As a matter of fact, we keept the brains, the muscle, the serums, and, right now are searching for neurotranmitters, cathecolamines, andocanabioniods and, cytokines, to, explain the “bold effect” as well as molecularmechanisms associated to exercise. 

Q- Does the exercise intensity at the end of the tenth week equate to light, moderate or heavy exercise? The type of intensity can directly influence the results and it is important to describe this sentence in the text.

A. It has been clarified in the text as suggested

Q- Authors should describe exercise protocol in a table.

A. It has been corrected as suggested

Q- on page 149 delete a comma.

A. It has been corrected as suggested

Q- On page 173, were the animals sacrificed 1 or 2 days after the end of the exercise?

A. It has been corrected as suggested, making it clear in the text. The animals were sacrificed right afeter the last test

Q- On page 343 the authors describe "medium intensity resistance exercise". Would it be a resistance or aerobic exercise? How do the authors know that indensity is medium?

A. Thank you. It has been clarifiedin the text

Q- On page 350 delete the word "that" before "underwent"

A. It has been corrected as suggested

Q.- Authors should discuss and investigate clinical studies of the effect on reducing the anti-inflammatory immune response

A. Thank you. That it is indeed the second phase of our studies. To include an antigenic challenge to search if, immune system activity is boosted or compromized in response to chronic exercise. In regards to clinic studies, it i suite hard, since only epidemiologic and correlative studies can be made, but, we will try out to partner to Hospitals and clinics to search that. 

Q.- On page 396 change "endo-cannabinoid" to endocannabinoid.

A. It has been corrected as suggested

Q- As the authors could explain the increased expression of CB1 receptors in same cells investigated, whereas CB2 receptors are expressed in most immune system?

A. That it is a hard question to answer. However, it has been widely suggested that CB1 type receptors, are expresed more in endocrine related tisuues, while, CB2, in other cell types, including immune cells. However, most of the studies on cannabinod receptiors are done either by western blott or immunocitochemistry, while we use flow cytometry (which it is more much reliable and accurate, since the expression it is at cell level)

---

## [Decision Letter · Decision Letter 1]

16 Oct 2019

PONE-D-19-20066R1

Cellular innate and adaptive immunity are affected by chronic exercise

PLOS ONE

Dear Dr. Morales-Montor,

Thank you for submitting your manuscript to PLOS ONE. After careful consideration, we feel that it has merit but does not fully meet PLOS ONE’s publication criteria as it currently stands. Therefore, we invite you to submit a revised version of the manuscript that addresses the points raised during the review process.

We would appreciate receiving your revised manuscript by Nov 30 2019 11:59PM. To enhance the reproducibility of your results, we recommend that if applicable you deposit your laboratory protocols in protocols.io, where a protocol can be assigned its own identifier (DOI) such that it can be cited independently in the future. For instructions see: http://journals.plos.org/plosone/s/submission-guidelines#loc-laboratory-protocols

We look forward to receiving your revised manuscript.

Kind regards,

Junpeng Wang, Ph.D

Academic Editor

PLOS ONE

Reviewers' comments:

Reviewer's Responses to Questions

**Comments to the Author**

1. If the authors have adequately addressed your comments raised in a previous round of review and you feel that this manuscript is now acceptable for publication, you may indicate that here to bypass the “Comments to the Author” section, enter your conflict of interest statement in the “Confidential to Editor” section, and submit your "Accept" recommendation.

Reviewer #2: (No Response)

Reviewer #3: All comments have been addressed

2. Is the manuscript technically sound, and do the data support the conclusions?

Reviewer #2: Yes

Reviewer #3: Yes

3. Has the statistical analysis been performed appropriately and rigorously? 

Reviewer #2: Yes

Reviewer #3: Yes

4. Have the authors made all data underlying the findings in their manuscript fully available?

Reviewer #2: Yes

Reviewer #3: Yes

5. Is the manuscript presented in an intelligible fashion and written in standard English?

Reviewer #2: Yes

Reviewer #3: Yes

6. Review Comments to the Author

Reviewer #2: 1- The authors have summarized the title, but the main tissue (endocannabinoid system) was excluded.

2- Ok!

3- At the end of the introduction the authors describe hypothesis, as suggested, but now the study objectives are omitted.

4- When asked the authors whether exercise activates or inhibits the immune system for the purpose of describing in the introduction, they replied that there is no study, but I suggest checking the review and describing any study addressed by it (10.3389/fimmu.2018.00648).

5- About weight of the animals, its ok!

6- About electrical stimulus, its ok!

7- The table 1 figure legend need to describe each abbreviation. Ok about requested table 1.

8- The last question about cb2 receivers should be included in the discussion. It is important to note that the results of the literature on western blot receptor-enhanced cb2 receptors and immunofluorescence are relevant and cannot be lowered compared to cytometry, even though cytometry has good accuracy.

Reviewer #3: The main objective of this study was to evaluate the effect of chronic treadmill exercise on lymphocytes, NK cells and macrophages of male rats investigating possible alterations of cannabinoid receptors in this process. The study is very interesting and important to better understanding of chronic exercise in immune cells. However, there are some minor points that need to be addressed.

1. The title need to be changed to indicate the important effects on cannabinoids receptors.

2. The purpose of the study and the main hypothesis is not clear in the abstract.

3. The identification of CD11b is sufficiently to identify macrophages?

4. The figures quality need to be improved.

7. PLOS authors have the option to publish the peer review history of their article (what does this mean?). If published, this will include your full peer review and any attached files.

Reviewer #2: No

Reviewer #3: No

---

## [Author Response · Author response to Decision Letter 1]

31 Oct 2019

Reviewer #2: 

1- The authors have summarized the title, but the main tissue (endocannabinoid system) was excluded.

Reply: Thank you for the observation, the title has now been changed to “Chronic exercise modulates the cellular immunity and its cannabinoid receptors expression”, including the cannabinoid system in the title once again.

2- Ok!

3- At the end of the introduction the authors describe hypothesis, as suggested, but now the study objectives are omitted.

Reply: We had already included the objectives of our work before the hypotesis. nontheless, in order to make it clearer we have changed it to: 

“Altogether, The objective of our study is to explore the long-term changes that chronic exercise (CE) produces in the proportion of splenocytes from the adaptive and innate immunity, and to assess the effects that it has on their function (by performing proliferation tests and cytotoxicity test with total splenocytes in vitro), and finally to determine if the expression of CBR in these cells is affected by this model of exercise.”

4- When asked the authors whether exercise activates or inhibits the immune system for the purpose of describing in the introduction, they replied that there is no study, but I suggest checking the review and describing any study addressed by it (10.3389/fimmu.2018.00648).

Reply: we added recent information envolving the interaction between exercise, metabolic deseases and cancer, this published data even suggest its use as an alternative therapy. This addition can be found at the end of the first paraghaph of the introduction as previously requested.

About the interesting review suggested, some lines have been added on the introduction making reference to this work and its take homme message about a new interpretation of older data, and the implications it has on the understanding of the beneficial physiological changes that the organisms undoergoes during the physical activity.

5- About weight of the animals, its ok!

6- About electrical stimulus, its ok!

7- The table 1 figure legend need to describe each abbreviation. Ok about requested table 1.

Reply: the abbreviations have been added.

8- The last question about cb2 receivers should be included in the discussion. It is important to note that the results of the literature on western blot receptor-enhanced cb2 receptors and immunofluorescence are relevant and cannot be lowered compared to cytometry, even though cytometry has good accuracy.

Reply: According to the question previously made, we have added a couple of lines in the cannabinoid receptors section (in the dicussion), where we explain the need for a double and independent staining for every cell suspention aliquot in order to assess the exprssion of each CBR due to the shared epitope for the secondary antibody.

We have also added some references to previous data regarding the assesment of CBr through different techniques, like western blot and ummunohistochemestry, highlighting our position with respect to the validity of such techniques and the information they provide.

Reviewer #3: The main objective of this study was to evaluate the effect of chronic treadmill exercise on lymphocytes, NK cells and macrophages of male rats investigating possible alterations of cannabinoid receptors in this process. The study is very interesting and important to better understanding of chronic exercise in immune cells. However, there are some minor points that need to be addressed.

1. The title need to be changed to indicate the important effects on cannabinoids receptors.

Reply: Thanks for the observation, we have includded the cannabinoid receptors on the title.

2. The purpose of the study and the main hypothesis is not clear in the abstract.

Reply: the objectves and hypohesis have been added as requested.

3. The identification of CD11b is sufficiently to identify macrophages?

Reply: This is an important observation. 

The expression of CD11b is not exclusive of macrophages, it is shared among different cells with myeloid origin, but once we restrict their identification to big and coplex cells, we obtain a reasonably adecuate identification of this cell subset. 

This aclaration has been added in the section of “Materials and methods” and the subsection “Flow Cytometry”.

4. The figures quality need to be improved.

Reply: the images lost quality when converted to .tiff the first time, making some lines disapear or blurry. That has been fixed. Now they have been converted to eps. format which alowed us to improve the resolution to 500 dpis, whitout exceeding the 10mb limit for the upload.

---

## [Editor Report · Decision Letter 2]

4 Nov 2019

Chronic exercise modulates the cellular immunity and its cannabinoid receptors expression

PONE-D-19-20066R2

Dear Dr. Morales-Montor,

We are pleased to inform you that your manuscript has been judged scientifically suitable for publication and will be formally accepted for publication once it complies with all outstanding technical requirements.

With kind regards,

Junpeng Wang, Ph.D

Academic Editor

PLOS ONE
---

## [Editor Report · Acceptance letter]

11 Nov 2019

PONE-D-19-20066R2 

Chronic exercise modulates the cellular immunity and its cannabinoid receptors expression 

Dear Dr. Morales-Montor:

I am pleased to inform you that your manuscript has been deemed suitable for publication in PLOS ONE. Congratulations! Your manuscript is now with our production department. 

With kind regards,

on behalf of

Dr. Junpeng Wang 

Academic Editor

PLOS ONE